# Effects of a Digital Health Literacy Intervention on Porcine Cysticercosis Prevalence and Associated Household Practices in Iringa District, Tanzania

**DOI:** 10.3390/pathogens12010107

**Published:** 2023-01-09

**Authors:** Flora Kajuna, Beda Mwang’onde, Christine Holst, Bernard Ngowi, Felix Sukums, Josef Noll, Andrea S. Winkler, Helena Ngowi

**Affiliations:** 1The College of Veterinary Medicine and Biomedical Sciences, Sokoine University of Agriculture, Morogoro P.O. Box 3021, Tanzania; 2The Livestock Training Agency, Ministry of Livestock Development and Fisheries, Dar es Salaam P.O. Box 3101, Tanzania; 3The Department of Biosciences, College of Natural and Applied Sciences, Sokoine University of Agriculture, Morogoro P.O. Box 3038, Tanzania; 4Centre for Global Health, Institute of Health and Society, University of Oslo, 0373 Oslo, Norway; 5Muhimbili Medical Research Centre, National Institute for Medical Research, Dar es Salaam P.O. Box 3436, Tanzania; 6Muhimbili College of Health and Allied Sciences, University of Dar es Salaam, Mbeya P.O. Box 608, Tanzania; 7The Directorate of Information and Communication Technology, Muhimbili University of Health and Allied Sciences, Dar es Salaam P.O. Box 65001, Tanzania; 8Basic Internet Foundation, Gunnar Randers vei 19, 2007 Kjeller, Norway; 9Centre for Global Health, Department of Technology Systems, University of Oslo, 0373 Oslo, Norway; 10Center for Global Health, Department of Neurology, Technical University of Munich, 81675 Munich, Germany

**Keywords:** digital, health, control, endemic, cysticercosis

## Abstract

Digital health is considered an opportunity to engage a wider community in disease control for public health. It has been used in healthcare consultation, in medical treatments and in reporting emergencies. The current study developed digital health literacy content for public health education and assessed its effects on porcine cysticercosis prevalence, pig-keeping style and pig pen and latrine qualities. The intervention was designed and evaluated on the prevention and control of porcine cysticercosis in the Iringa District of southern Tanzania. A quasi-controlled field trial with pre-intervention and post-intervention assessments of porcine cysticercosis, pig-keeping style and pig pen and latrine qualities was conducted. A baseline cross-sectional study was followed immediately by digital health literacy intervention, which comprised educational messages on porcine cysticercosis shown on computer tablets or smartphones. Free internet access supported unsupervised community access. The 25-month post-intervention assessments revealed significantly increased pig confinement (20.1%) (*p* = 0.026) and pig pen quality (16.2%) (*p* = 0.025). However, the quality of household latrines (*p* = 0.453) was not improved, nor was there any significant effect on the prevalence of porcine cysticercosis (*p* = 0.231). The digital health literacy intervention suggests a strategy for wider and sustainable dissemination of educational messages for *Taenia solium* infection control.

## 1. Introduction

Worldwide public health education is used to improve the control of diseases by increasing literacy and providing practical guidelines for the control of targeted diseases [1]. Digital technology has provided an opportunity for the quick and wide dissemination of education, including health education, and hence enhances disease management and control [2,3]. *Taenia solium* cysticercosis is one of the important but neglected zoonotic diseases endemic in Africa, Asia and Latin America, with considerable public health and economic impacts [4,5]. The disease is detrimental to human health, since it causes neurocysticercosis that reduces life quality [6,7], and it affects the pig industry by disrupting pork quality, hence reducing income from pig business while creating a threat to public health [8,9]. 

Tanzania is one of the African countries highly affected by *T. solium* cysticercosis in humans and pigs. The disease is endemic in many regions of the country, including the northern, central and southern highlands [10,11]. Different intervention strategies have been applied to control *T. solium*, with health education suggested as an important component in any control strategy for *T. solium* infections [12,13,14]. Digital technology has been suggested as a quicker and more convenient means of delivering health education to the public to enhance the control of *T. solium* cysticercosis [15,16,17]. The technology has improved the knowledge and altitude of the awareness of the control of *T. solium* cysticercosis [4], though the intervention effect based on actual practices for disease management is lacking.

A project named “Non-discriminating Access for Digital Inclusion (the DIGI project)” in Tanzania, funded by the Norwegian Research Council, developed health education messages for *T. solium* infection control and disseminated them to selected pilot communities in southern Tanzania, through showing videos and the provision of free internet access. The implemented program allowed the local communities (individually or in groups) to learn the epidemiology, prevention and control of *T. solium* infections. This study aimed at assessing the effects of the digital health intervention on the prevalence of *T. solium* cysticercosis in pigs and favourable practices for the control of the infection. 

## 2. Materials and Methods

### 2.1. Study Area

This study was conducted in three wards, namely Izazi, Migoli and Mlowa, involving nine villages. The study sites are within the Iringa District in the southern highlands of Tanzania. In 2012, the Iringa District was estimated to have 254,032 people [18], and the pig population was estimated to be 35,065 out of the total 241,829 pigs accounted for in the Iringa region [18], which is among the southern highland regions producing pigs in Tanzania. The district is located between 7° 05′ and 36° 32′ south and 33° 47′–36° 32′ east with an area of 20,576 km^2^. The Iringa District lies at an altitude of 475 meters above sea level with high peaks of 2981 meters above sea level. Rainfall reaches up to 704 mm per year, and the cold season’s temperature falls below 15 °C. The wet season occurs from November to May with a peak in January [19]. The district predominantly produces pigs on a small scale, and contributes about 15 per cent of the pig population in Tanzania [18]. Although the pig production industry is known to improve livelihoods [20], the industry has been affected by porcine cysticercosis [9]. The present study was conducted from April 2019 to July 2021. It is important to note in this study that the study site had sanitation campaigns in the year 2020, which emphasized the construction and use of latrines using sanitary methods among households. In addition, in the same year, district and village leaders enforced by-laws which insisted on indoor pig-keeping. In addition, the COVID-19 pandemic emerged in Tanzania around February 2020, which stimulated sanitation (especially handwashing) that was highly emphasized.

### 2.2. Study Design

This was a quasi-field trial with pre-intervention and post-assessment of porcine cysticercosis and related household practices. The study involved a baseline assessment of the prevalence of porcine cysticercosis, pig-keeping style and pig pen quality and the latrine quality assessment, followed by a digital health intervention (started immediately after the baseline to 25 months) and finally a follow-up survey similar to the baseline study (25 months after the intervention). Pig-keeping style was defined as poor for scavenging or tethering yearly and good for confinement all year. Poor pig pen quality was defined as pens that cannot support total animal confinement and animal welfare, while good pig pens were those that can support total animal confinement and animal welfare all year. In addition, latrine quality was defined as poor when it discouraged use and had an unrestricted entrance, but was defined as good when it encouraged use and had a restricted non-human entrance. The digital health intervention assessment was for the evaluation of the intervention effect. Post-intervention data were collected in July 2021.

### 2.3. Sample Size and Sampling

The sample size for pigs was calculated using the formula *n* = Z^2^PQ/L^2^ [21], where “*n*” is the required sample size, “P” is the known or estimated prevalence of the desired factor (in this case porcine cysticercosis), “Q” is 1-P, which is a proportion free of the factor, and “L” represented acceptable estimation error. “Z” is the z-score for a confidence level, which was 1.96, as we preferred a 95 per cent confidence level. Consequently, “L” was 5 per cent. Based on previous studies in neighbouring districts of Mbozi and Mbeya, “P” was estimated at 33 per cent [22]. Thus, 339 pigs were required for the study. About 346 pigs were examined from 88 households at baseline. During the post-intervention assessment, a total of 298 pigs from 70 households who continued keeping pigs underwent examination.

### 2.4. Data Collection

Biodata and a brief history were collected for each study pig, including the likelihood of the female pig is pregnant. An approximately 5 mL blood sample was collected through the external jugular vein into a plain vacutainer tube [23] and allowed to clot. The clotted blood samples were centrifuged at 2000 revolutions per minute (rpm) for 3 min. Each serum sample was kept in a 2 mL cryogenic tube and stored at −20℃ before analysis. Piglets younger than two months and sows with detectable pregnancy were excluded to reduce chances of stress which could have adverse health effects. Using a commercial Enzyme-Linked Immunosorbent Assay (ELISA) kit [24] as per the manufacturer’s (apDia bvba, Raadsherenstraat 3, 2300 Turnhout, Belgium) instruction, the seropositivity status of each pig was obtained.

Parallel to pig sampling was the observational study that aimed at assessing household practices related to pig management, sanitation and hygiene. Pig-keeping style and pig pen and latrine qualities were examined to determine good or poor practices. A checklist of targeted factors guided the observational study. Interviews and observational data were collected in Kobo toolbox software installed on Android tablets.

### 2.5. The Digital Health Literacy Intervention

The digital health literacy intervention consisted of two phases. The first phase was from April to May/June 2019, soon after baseline data collection. This phase involved research assistants showing a video to some randomly selected household members of the intervention wards (Izazi and Migoli). The video consisted of animated health messages on *T. solium* taeniasis/cysticercosis. The tablets with the video were left in the ward offices for more voluntary learning in the villages. The second phase was from November 2019 to September 2020, which involved the provision of free information and internet access in an InfoSpot (a centre with free wireless access to a health platform and the internet). In the InfoSpot, the community had unsupervised access to health education in the form of texts, pictures, animations and quizzes. Viewers were expected to access the educational content using personal smartphones or the DIGI project tablets available in the InfoSpots and ward offices in the intervention wards.

### 2.6. Intervention Allocation

Izazi and Migoli Wards were assigned to the intervention, while Mlowa Ward served as a control because of its isolated location, which was considered favourable to reduce chances of information communication and contamination.

### 2.7. Post-Intervention Data Collection

The post-intervention data were collected from the same households examined during the baseline still keeping pigs. The same data collection procedures as those used in the baseline study were used during the post-intervention study.

### 2.8. Data Analysis

Data were entered into a Microsoft Excel spreadsheet and analysed in STATA 15 to assess the effect of the digital health intervention on porcine cysticercosis prevalence and the three household practices: pig-keeping style, the quality of a constructed pig pen and the quality of a household latrine. This analysis involved only households that participated in both the baseline and the post-intervention studies, hereby referred to as “full participants”. To avoid the effect of clustering at the household level, although several pigs were examined in some households based on availability, the intervention effect was analysed at the household level by considering a household positive if it had at least one infected pig at the visitation time. First, McNemar’s Chi-Square tests were used to see if, within a treatment group (intervention or control) there was any significant difference in the prevalence of at least one infected pig in a household between the baseline and post-intervention periods. Therefore, baseline and post-intervention data within a group were considered as paired since the same households were examined between the two periods. However, a different pig population was most likely examined post-intervention because of the long time difference between the two periods (approximately 25 months), in which most baseline pigs would have been sold. Thus, the presence of at least one infected pig in a household was considered a household factor. McNemar’s Chi-Square test was computed in STATA 15 using the command *mcc*. To analyse the effect of the intervention on the studied variables, the intervention and control households were considered independent. Therefore, the STATA command *prtest* was used to analyse the effect of the digital health intervention on the prevalence of at least one case of porcine cysticercosis in the household. In addition, the data at the pig level were analysed using the Wilcoxon signed-rank test for within-treatment group changes from baseline and Wilcoxon rank-sum test for between-treatment group differences (intervention effects). Pig-keeping style, pig pen quality and latrine quality were the analysed household practices. Although the practices were rated into three levels, good, fair and poor (or similar terms), in the analysis of the intervention effect, all variables were dichotomised into good or poor by merging the poor and fair into poor. The intervention effects on these factors were analysed using the same statistical procedures used in analysing the intervention effect on porcine cysticercosis at the household level described above.

## 3. Results

### 3.1. General Results

This study included 88 households during the baseline (48 intervention and 40 control). At the post-intervention assessment, 18 households (16 intervention and 2 control) could not be followed up because they were no longer keeping pigs. Therefore, 70 households (32 intervention and 38 control) participated in both assessments, and hence, they were analysed for the intervention effect. The baseline prevalence of porcine cysticercosis and its associated risk factors have been reported elsewhere [25].

### 3.2. The Effect of Digital Health Literacy Intervention on The Prevalence of Porcine Cysticercosis

The difference in the pig-level prevalence of porcine cysticercosis between the intervention and control groups is shown in Table 1. There was no significant difference in the pig-level prevalence of porcine cysticercosis between the baseline and post-intervention periods in the intervention group, despite a slight decrease during the post-intervention period (*p* = 0.365). On the other hand, there was a significant decrease in the pig-level prevalence of porcine cysticercosis during the post-intervention period in the control group (*p* = 0.026). The non-parametric analysis of the pig-level data using the Wilcoxon signed-rank test for within-treatment group changes from baseline and Wilcoxon rank-sum test for between-treatment group differences did not find any statistical significance (*p* = 0.6988, 0.1175 and 0.5184, respectively).

Having found no significant difference in the prevalence of porcine cysticercosis between the two intervention wards (Izazi and Migoli), in the subsequent analysis, the two wards were analysed in combination. Although the percentage of households with at least one positive case increased from 50 to 53.1 per cent in the intervention group, this increase was not statistically significant (*p* = 0.7963, *n* =32). Similarly, the decrease in the percentage of households with at least one infected pig from 39.5 to 28.9 per cent in the control group was not statistically significant (*p* = 0.3938, *n* = 38) (Table 2).

This analysis found no statistical difference that could be attributed to the intervention (*p* = 0.2310), as while some households changed from disease to no disease status in both groups, a reasonable number of households changed from no disease to disease status, making it difficult to rule out the role of chance.

### 3.3. Effect of The Digital Health Literacy Intervention on Household Practices Related to Porcine Cysticercosis

Intervention effects on a change from baseline levels of the pig-keeping style, pig pen quality and latrine quality are shown in Table 3. The within-treatment group comparison between the baseline and post-intervention periods using McNemar’s Chi-Square test found a significant change from scavenging/tethering to confinement of pigs in both the intervention and control households (*p* = 0.000 in both). There were no significant differences between the baseline and post-intervention periods on pig pen quality (*p* = 0.263 and 1.000, respectively) or latrine quality, in either of the treatment groups (*p* = 0.146 and 0.064, respectively). The comparison between groups revealed a significant change in good practices of keeping pigs under confinement and constructing good-quality pig pens attributable to the intervention (*p* = 0.026 and 0.025, respectively).

## 4. Discussion

This study has, for the first time, evaluated the effect of a digital health literacy intervention on the prevalence of porcine cysticercosis and three associated actual practices in an endemic setting in Tanzania. A study by [4] reported improved knowledge and attitudes towards the control of porcine cysticercosis through digital health education in Tanzania. Such kind of education has not been found to predict the intervention effect on pig management practices and with no disease magnitude. This pilot study has shown promising results towards the use of digital technology to enhance the dissemination of health education messages to reduce endemic diseases, particularly neglected tropical diseases in sub-Saharan Africa, as the intervention improved some key practices necessary for the control of *T. solium* infections.

The digital health literacy intervention on *T. solium* infection control caused significant improvement in confinement and animal welfare, as well as a marked increase in pig confinement. These are important measures to prevent pigs from acquiring infections from contaminated environments. This positive finding suggests that knowledge has influenced practices positively. A study by [12] found no significant improvement in pig confinement approximately 10–12 months following a community-based face-to-face health education intervention, despite improvement in knowledge and attitudes. The authors deduced that the time was too short to observe any change in practices based on the Diffusion of Innovation Theory [26]. The remarkable improvement in pig house quality and pig confinement observed in the present study could be because of the longer duration of 25 months post-intervention compared to the previous study [12]. It could also be attributed to the easiness, and the strength in conveying digitally enabled health messages and animations used to educate the community on practising good pig husbandly for porcine cysticercosis control. The findings of this study are consistent with a similar study in Mozambique, which revealed changes to good pig-keeping style and construction of good-quality pig pens through education [27]. 

The digital health literacy intervention did not improve the quality of household latrines. This suggests that some behaviour changes may need more time to allow knowledge synthesis to practical outcomes [28,29], as they may be tied to cultural norms, making them difficult to change within a short time. Studies elsewhere have indicated that changes in health management practices in communities require regular education and the provision of enabling skills for the implementation of essential public health measures [30]. Skills and/or guidance in constructing a pig pen were not provided in the current education program, which could have a large effect on the quality of the construction of pig pens in the study area.

The digital health literacy intervention did not have a significant effect on the prevalence of porcine cysticercosis. The lack of any effect from the intervention could partly be attributed to the lack of improvement in household latrine quality. Despite increased confinement of pigs observed during our snapshot visit to the households, such pigs may still be at risk of infection due to poor latrines and confinement is not permanent practice. It was reported that some farmers do release their pigs to access field crop leftovers post-harvest, especially during the dry season [31]. This may allow pigs to access human faecal material, which may sometimes contain *T. solium* eggs. The general decline in porcine cysticercosis prevalence post-intervention in both the intervention and control households observed in this study implies that the decline cannot be related to the intervention. Still, rather it could be a result of other factors that occurred in the study areas, which may include sanitation campaigns carried out in the year 2020 that emphasised the construction and use of latrines using sanitary methods among households. In addition, in the same year, district and village leaders enforced by-laws that insist on indoor pig-keeping. Another factor was the COVID-19 pandemic that emerged in Tanzania in February 2020, in which sanitation (especially hand washing) was highly emphasised. In Burkina Faso and Tanzania, porcine cysticercosis has been reported as more prevalent in free-roaming than among confined pigs, especially during cold weather or the rainy season [31,32]. Pigs scavenging on farm leftovers, practised during the dry season, likely exposed the pigs to porcine cysticercosis infection that becomes detectable two to three months later, a period falling in the following rainy season. This could account for possible factors that led to a higher prevalence of porcine cysticercosis at baseline (end of rainy season) compared to the post-intervention period (dry season). Presence of tall grasses and bushes after rainy season create convenient environment away from home for open defecation [33], hence posing a risk to roaming and even confined pigs feeding on cut grass. Grown shrubby and shade trees provide cool weather conditions for pig habitation far away from their homes [34], widening the pigs’ chances of coming into contact with *T. solium* eggs in contaminated environments. Running water in the rainy season increases the chances of environmental contamination by dispersing materials, including worm eggs, from contaminated environments. However, the decline in the prevalence post-intervention in both the intervention and control wards was not statistically significant. The pig producers were not restricted to selling or slaughtering their pigs, and the fact that the post-intervention sampling was conducted two years from the time of the baseline study makes it likely that the pigs examined during the post-intervention were different from the ones examined during the baseline, thus possibly reflecting new cases (incidents) of porcine cysticercosis.

## 5. Conclusions

This study revealed a significant improvement in pig pen quality and increased pig confinement attributable to the digital health literacy intervention in Izazi and Migoli Wards of the Iringa District of southern Tanzania. This is an important step forward towards the control of *T. solium* infections. The intervention did not improve the quality of household latrines, nor did it reduce the prevalence of porcine cysticercosis. This may be explained by the two factors being interwoven and that more time is needed to see changes resulting from an education intervention. The authors recommend the digital health literacy intervention as a strategy for wider and sustainable dissemination of educational messages for the control of *T. solium* infections and other public health problems. Sustainability of the digital health intervention remains a challenge to solve, possibly between the public sector (such as the ministry of health) and private sector, in particular mobile phone operators, to enable poor rural communities to afford the costs associated with internet access.

## Figures and Tables

**Table 1 pathogens-12-00107-t001:** Effect of the digital health literacy intervention on porcine cysticercosis prevalence at pig level in Iringa Rural District, Tanzania, 2019–2021.

Ward	Group	Baseline Study			Post-Intervention Study			Difference	Statistical Significance
		Tested	+ve	Prev. (%)	Tested	+ve	Prev. (%)	Post interv. and Baseline	*p*-Value
Izazi	intervention	56	12	21.4	69	10	14.5	−6.9	0.311
Migoli	intervention	89	18	20.2	110	20	18.2	−2.0	0.715
Izazi and Migoli	intervention	145	30	20.7	179	30	16.8	−3.9	0.365
Mlowa	control	114	26	22.8	119	14	11.8	−11.0	0.026

**Table 2 pathogens-12-00107-t002:** Effect of a digital health literacy intervention on the prevalence of at least one case of porcine cysticercosis in a household in Iringa Rural District, Tanzania, 2019–2021.

Group	Households Examined (a)	Households Changing from Positive to Negative After Intervention (b)	Households Changing from Negative to Positive After Intervention (c)	Net Change in Status After Intervention (d = c − b)	Percentage Net Change in Status After Intervention (d/a × 100)	Difference in Change Between Control and Intervention in % (and 95% CI)	Significance of The Change in Status Between Groups (*p* Value)
Control	38	13	9	4	10.5	7.4 (−4.1, 18.9)	0.231
Intervention	32	7	8	1	3.1		

**Table 3 pathogens-12-00107-t003:** Effect of the digital health literacy intervention on household pig-keeping style, pig pen quality and latrine quality in Iringa Rural District, Tanzania, 2019–2021.

Group	Households Examined (a)	Households Changing from Poor to Good Practice After Intervention (b)	Households Changing from Good to Poor Practice After Intervention (c)	Net Change in Status After Intervention(d = c − b)	Percentage Net Change in Status After Intervention (d/a × 100)	Difference in Change Between Control and Intervention in % (and 95% CI)	Significance of the Change in Status Between Groups (*p* Value)
**Pig-keeping style**							
Control	38	29	1	28	73.7	20.1 (3.7, 36.4)	0.026
Intervention	32	30	0	30	93.8		
**Pig pen quality**							
Control	38	10	9	1	2.6	16.2 (16.7, 30.6)	0.025
Intervention	32	13	7	6	18.8		
**Latrine quality**							
Control	38	7	17	10	26.3	7.5 (−27.0, 11.9)	0.453
Intervention	32	3	9	6	18.8		

## Data Availability

Not applicable.

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
