# Peer review of "Effects of a Digital Health Literacy Intervention on Porcine Cysticercosis Prevalence and Associated Household Practices in Iringa District, Tanzania"

_pathogens, 2023, doi:10.3390/pathogens12010107_

Round 1
Reviewer 1 Report
The MS “Effects of a Digital Health Literacy Intervention on Porcine Cysticercosis Prevalence and Associated Household Practises in Iringa District, Tanzania” provides interesting information that is worth publishing. I suggest some modifications that will makes easier to read and follow this study:
Abstract:
> “Digital” not in bold.
>Change “associated house-hold factors” by the three factors that you analyzed, so the abstract will be more informative
>Include the percentatges of increase/decrease for pig confinements and the quality of pig pens.
> “Digital health literacy intervention suggested a strategy for wider and sustainable dissemination of
educational messages for Taenia solium infections.’ Remove “and other public health problems control” as is speculative, you did not proof this part in your study
Introduction
>Other similar studies of Digital Health Literacy and porcine cysticercosis” need to be presented in the Introduction, so we know what has been done previously (these studies are in the discussion, but reader need to know the current status.
>Taenia solium can be abbreviated to T. solium in all the MS (except the first time you cite).
Data collection
> “Using a commercial Enzyme-Linked Immunosorbent Assay (ELISA) kit” please detail the company you used.
> “Parallel to pig sampling was the observational study that aimed at assessing house-
hold practices related to pig management, sanitary and hygiene.” Add the three factors that you analyzed to make sentence more concrete “…sanitary and hygiene: pig keeping style, ……”.
Intervention allocation
> “…Izazi and Migoli Wards were assigned to the intervention while Mlowa Ward served as a control because of its isolated location, which was considered favourable to reduce chances of information communication and contamination”
This has much more sense to be included in “study area”
Data analysis
> “….and related household practices” change by more concrete sentence “….and the three household practices: ……..”
> “…ere the three main household practices analysed” I do not see you analyze other household practices, so reformulate the sentence.
> “Table 1. presents the difference in the pig-level prevalence of porcine cysticercosis between the intervention and control groups.” Improve the sentence as “The difference in the pig-level prevalence of porcine cysticercosis between the intervention and control groups are showed in Table 1”.
>Do the same in “Table 3. presents the analysis of intervention effects….”
>In table 3 the definition of “poor” and “Good” should be included in material and methods, so the table will be easier to read. Also detail in material and methods what you consider is good animal welfare.
For example: Pig-keeping style
(Poor = Scavenging / tethering;
Good = Confinement)
>In table: Should be Pig-keeping style
Poor
Good
>An in text detail what means poor and what means good
Discussion
>“A similar intervention study in Tanzania assessed the intervention effects on knowledge and attitudes [4].” Looks this study is in fact a review, reformulate the sentence.
>“…and associated actual practices” change by “ and three associated actual practices”
> “The digital health literacy intervention on Taenia solium infection control caused sig-nificant improvement in pig pen quality” change “pig pen quality” to be more concrete “confinement and animal welfare”
> “Despite increased confinement of pigs observed during our snapshot visit to the households, such pigs may still be at risk of infection due to poor latrines as the confinement might not be a permanent practice, especially during the dry season when some farmers release their pigs to access field crop leftovers of post-harvest.”
>Do you have your information? As when you divide in confinement and not confinement, the first category is not really true as probably a part of year in not confinement. So in material and methods “confinement” means that in apart of the year pigs are confined but not all year, so make it clear. In case you have this information on your cases: all year confined vs not all year confined will be an useful information”
> “Rather it could be a result of other factors that occurred in the study areas, which may include sanitation campaigns carried out in the year 2020, which emphasized the construction and use of latrines in sanitary ways among households. Also, in the same year, district and village leaders enforced by-laws that insist on indoor pigs keeping. Another factor was the COVID-19 pandemic that emerged in Tanzania in February 2020 in which sanitation (especially handwashing) was highly emphasized.”
This is more appropriate in Study area to give context in your study (also you can keep in the Discussion but need to be introduced in Study area as is an important characteristic to understand the frame of your study.
References
>In some references Taenia solium is not in italics Taenia solium.
I feel a minor English revision will be useful as for example some sentences are too short and can be connected with another short sentence to make an easier-to-read text.
For example : “Z” is the z-score for a confidence level.
This was 1.96 as we preferred a 95 per cent confidence level.
This can be a single sentence.
Author Response
Reviewer 1.
Point: The MS “Effects of a Digital Health Literacy Intervention on Porcine Cysticercosis Prevalence and Associated Household Practises in Iringa District, Tanzania” provides interesting information that is worth publishing. I suggest some modifications that will makes easier to read and follow this study:
Response: Thank you for impressing comment about information we provide in the manuscript. The suggested modifications are made for the easier read and follow the study.
Points on Abstract:
> “Digital” not in bold.
Response: The word “Digital” has been un-bolded
>Change “associated house-hold factors” by the three factors that you analyzed, so the abstract will be more informative
Response: The “associated house-hold factors” have been replaced with pig-keeping style, pig-pen and latrine qualities.
>Include the percentages of increase/decrease for pig confinements and the quality of pig pens.
Response: The percentages for increased pig confinements and quality of pig pens have been placed. The sentence is reading as “The 25 months post-intervention assessments revealed significantly increased 20.1% pig confinements (p = 0.026) and 16.2% quality pig pens (p = 0.025)”.
> “Digital health literacy intervention suggested a strategy for wider and sustainable dissemination of educational messages for Taenia solium infections.’ Remove “and other public health problems control” as is speculative, you did not proof this part in your study
Response: The sentence “and other public health problems control” has been removed as suggested
Points on Introduction
>Other similar studies of Digital Health Literacy and porcine cysticercosis” need to be presented in the Introduction, so we know what has been done previously (these studies are in the discussion, but reader need to know the current status.
Response: The Digital Health Literacy and porcine cysticercosis has been presented as, “The technology improved knowledge and altitude for the awareness on the control of Taenia solium cysticercosis [4]. Though intervention effect based on actual practices for the disease management was lacking”. Page 2, paragraph 2.
>Taenia solium can be abbreviated to T. solium in all the MS (except the first time you cite).
Response: The Taenia solium has been abbreviated to T. solium in the text except for the first time
Data collection
> “Using a commercial Enzyme-Linked Immunosorbent Assay (ELISA) kit” please detail the company you used.
Response: The Company apDia of Belgium has been detailed as suggested organized on a sentence as follow; “Using a commercial Enzyme-Linked Immunosorbent Assay (ELISA) kit [24] as per manufacturer’s (apDia, Belgium)” instruction. Page 3, paragraph 2.
> “Parallel to pig sampling was the observational study that aimed at assessing house-hold practices related to pig management, sanitary and hygiene.” Add the three factors that you analyzed to make sentence more concrete “…sanitary and hygiene: pig keeping style, ……”.
Response: The three analyzed factors have been added to make a sentence read as “Pig keeping style, pig-pen and latrine qualities were examined to determine for a good or poor practices”. Page 3, paragraph 3.
Intervention allocation
> “…Izazi and Migoli Wards were assigned to the intervention while Mlowa Ward served as a control because of its isolated location, which was considered favourable to reduce chances of information communication and contamination”. This has much more sense to be included in “study area”
Response: Sure, the study plan considered means of reducing chances of information contamination
Data analysis
> “….and related household practices” change by more concrete sentence “….and the three household practices: ……..”
Response: The sentence has been reorganized to read “…….. and the three household practices: the pig-keeping style, the quality of a constructed pigpen and the quality of a household-latrine”. Page 3, paragraph 7.
> “…ere the three main household practices analysed” I do not see you analyze other household practices, so reformulate the sentence.
Response: The sentence has been reorganized to read as “Pig-keeping style, pig-pen quality, and latrine quality were the analysed household practices. Page 4, paragraph 1.
> “Table 1. presents the difference in the pig-level prevalence of porcine cysticercosis between the intervention and control groups.” Improve the sentence as “The difference in the pig-level prevalence of porcine cysticercosis between the intervention and control groups are showed in Table 1”.
Response: The sentence has been reorganized to read as “The difference in the pig-level prevalence of porcine cysticercosis between the intervention and control groups are shown in Table 1”. Page 4, paragraph 3.
>Do the same in “Table 3. presents the analysis of intervention effects….”
Response: The sentence has been reorganized to read as “Intervention effects on a change from baseline levels of the pig keeping style, pig-pen quality and latrine quality are shown in Table 3”.
>In table 3 the definition of “poor” and “Good” should be included in material and methods, so the table will be easier to read. Also detail in material and methods what you consider is good animal welfare.
For example: Pig-keeping style (Poor = Scavenging / tethering; Good = Confinement)
Response: The definitions of “poor” and “Good” have been included in material and methods, as it read as “Pig keeping style was defined as poor for scavenging or tethering and good for the confined. The poor pig-pen quality was defined as the one that cannot support total animal confinement and animal welfare, while the good pig-pen was the one that can support total animal confinement and animal welfare. Also latrine quality was defined poor when it was discouraging the use and had unrestricted entrance, but it was defined as good when it encouraged on the use and had a restricted non-human entrance”. Page 4, paragraph 6.
>In table: Should be Pig-keeping style (Poor, Good)
Response: In table 3, the pig-keeping style has been kept as Poor, Good. Page 5 and 6.
>And in text detail what means poor and what means good
Response: in text it has been detailed on what is poor and what is good. Page 4, paragraph 6.
Discussion
>“A similar intervention study in Tanzania assessed the intervention effects on knowledge and attitudes [4].” Looks this study is in fact a review, reformulate the sentence.
Response: The sentence has been reformulated to read as “A study by [4] reported the improved knowledge and attitudes towards control of porcine cysticercosis through digital health education in Tanzania”. Page 6, paragraph 1.
>“…and associated actual practices” change by “ and three associated actual practices”
Response: The sentence has been changed to read as, “and three associated actual practices in an endemic setting of Tanzania”. Page 6, paragraph 1.
> “The digital health literacy intervention on Taenia solium infection control caused significant improvement in pig pen quality” change “pig pen quality” to be more concrete “confinement and animal welfare”
Response: The sentence has been improved to read “The digital health literacy intervention on T. solium infection control caused significant improvement in confinement and animal welfare……” Page 6, paragraph 2.
> “Despite increased confinement of pigs observed during our snapshot visit to the households, such pigs may still be at risk of infection due to poor latrines as the confinement might not be a permanent practice, especially during the dry season when some farmers release their pigs to access field crop leftovers of post-harvest.”
Response: Reorganization has been made to read “Despite increased confinement of pigs observed during our snapshot visit to the households, such pigs may still be at risk of infection due to poor latrines and if confinement might not be a permanent practice. It was reported some farmers do release their pigs to access field crop leftovers of post-harvest especially during the dry season [32]”. Page 6, paragraph 4.
>Do you have your information? As when you divide in confinement and not confinement, the first category is not really true as probably a part of year in not confinement. So in material and methods “confinement” means that in a part of the year pigs are confined but not all year, so make it clear. In case you have this information on your cases: all year confined vs not all year confined will be an useful information”
Response: In this study, pig confinement was meant to be the all year confinement. Page 2, paragraph 5.
> “Rather it could be a result of other factors that occurred in the study areas, which may include sanitation campaigns carried out in the year 2020, which emphasized the construction and use of latrines in sanitary ways among households. Also, in the same year, district and village leaders enforced by-laws that insisted on the indoor pigs keeping. Another factor was the COVID-19 pandemic that emerged in Tanzania in February 2020 in which sanitation (especially handwashing) was highly emphasized.”
This is more appropriate in Study area to give context in your study (also you can keep in the Discussion but need to be introduced in Study area as is an important characteristic to understand the frame of your study.
Response: The information has been organized in the study area as suggested. Page 2, paragraph 4.
References
>In some references Taenia solium is not in italics Taenia solium.
Response: The Taenia solium has been formatted in italics in the references
I feel a minor English revision will be useful as for example some sentences are too short and can be connected with another short sentence to make an easier-to-read text. For example: “Z” is the z-score for a confidence level. This was 1.96 as we preferred a 95 per cent confidence level. This can be a single sentence.
Response: The comment has been taken into consideration for English revision to improve the manuscript.
Reviewer 2 Report
This certainly interesting and generally well-written paper is certainly worthy of publication. The findings are from a certain perspective somewhat disappointing, but no less valid and worthy of dissemination. Only a few minor changes are suggested
Referee’s comments on: “Effects ofd a digital health literacy intervention on porcine cysticercosis prevalence and associated household practices in Iringa District, Tanzania” by Kajuna et al.
The lack of line numbers makes placing the following suggested corrections difficult!
Introduction
Page 2, top. It is probably unnecessary to say that T. solium cysticercosis is “detrimental to human health” and “a threat to public health”.
Lower down in the introduction, the meaning of the term “sensitisation” is a little obscure. A more precise term should be chosen.
Study area
When it says that there was a total of 241829 pigs in the region, does region refer to the whole country? If so better to state this categorically.
The meaning of “it has interfered with porcine cysticercosis” is also obscure – should be rephrased.
Page 4
The effect of digital health literacy…
“there was no any significant difference” – delete any, or is it meant to be “not any”.
Page 5. Table 3
The CI for the difference of change between control and intervention for pig pen quality is given as 16.7 to 30.6. The calculate figure given for the difference is outside this range, 16.2. In the next column over there appears to be a minus symbol before the p value of 0.025.
Page 6. Discussion
“actual practices with no disease magnitude”. Obscure meaning, please rephrase.
“It could also be attributed to the attractiveness of the …messages”. Is attractiveness the best word to use here?
Author Response
This certainly interesting and generally well-written paper is certainly worthy of publication. The findings are from a certain perspective somewhat disappointing, but no less valid and worthy of dissemination. Only a few minor changes are suggested
Response: Thank you for reviewing our work. Science written facts makes informative findings to the ongoing situation on a subject under the study.
Referee’s comments on: “Effects of a digital health literacy intervention on porcine cysticercosis prevalence and associated household practices in Iringa District, Tanzania” by Kajuna et al.
The lack of line numbers makes placing the following suggested corrections difficult!
Response: Thank you for reviewing our work. Sorry for such inconveniences.
Introduction
Page 2, top. It is probably unnecessary to say that T. solium cysticercosis is “detrimental to human health” and “a threat to public health”.
Response: The disease is detrimental to human health since it causes neurocysticercosis that reduces life quality. Page 2, paragraph 1.
Lower down in the introduction, the meaning of the term “sensitisation” is a little obscure. A more precise term should be chosen.
Response: The word “sensitization” has been replaced with the word “video show”. Page 2, paragraph 1.
Study area
when it says that there was a total of 241829 pigs in the region, does region refer to the whole country? If so better to state this categorically.
Response: The country has several regions, Iringa region is among the southern highlands regions producing pigs in Tanzania Page 2, paragraph 4.
The meaning of “it has interfered with porcine cysticercosis” is also obscure – should be rephrased.
Response: The sentence was rephrased to read “Although pig production industry is known to improve livelihoods [20], the industry has been interfered with porcine cysticercosis”. Page 2, paragraph 4.
Page 4
The effect of digital health literacy…
“there was no any significant difference” – delete any, or is it meant to be “not any”.
Response: Any has been deleted. Page 5, paragraph 1.
Page 5. Table 3
The CI for the difference of change between control and intervention for pig pen quality is given as 16.7 to 30.6. The calculate figure given for the difference is outside this range, 16.2. In the next column over there appears to be a minus symbol before the p value of 0.025.
Response: Difference of change between control and intervention for pig pen quality was obtained by 18.8 minus 2.6 that provided 16.2% at 95% CI. At the table 3.
Page 6. Discussion
“actual practices with no disease magnitude”. Obscure meaning, please rephrase.
Response: The sentence has been rephrased to, “Such kind of education has not been found to predict the intervention effect on the pig management practices and with no disease magnitude”. Page 6, paragraph 1.
“It could also be attributed to the attractiveness of the …messages”. Is attractiveness the best word to use here?
Response: It could also be attributed with the easiness and the strength in conveying digitally enabled health messages Page 6, paragraph 2.